# Recurrent/Metastatic Squamous Cell Carcinoma of the Head and Neck: A Big and Intriguing Challenge Which May Be Resolved by Integrated Treatments Combining Locoregional and Systemic Therapies

**DOI:** 10.3390/cancers13102371

**Published:** 2021-05-14

**Authors:** Franco Ionna, Paolo Bossi, Agostino Guida, Andrea Alberti, Paolo Muto, Giovanni Salzano, Alessandro Ottaiano, Fabio Maglitto, Davide Leopardo, Marco De Felice, Francesco Longo, Salvatore Tafuto, Giuseppina Della Vittoria Scarpati, Francesco Perri

**Affiliations:** 1Otolaryngology Unit, INT IRCCS Foundation G. Pascale, Naples. Via M. Semmola, 80131 Naples, Italy; f.ionna@istitutotumori.na.it (F.I.); giovanni.salzano@istitutotumori.na.it (G.S.); fabio.maglitto@istitutotumori.na.it (F.M.); 2Medical Oncology, Department of Medical and Surgical Specialties, Radiological Sciences and Public Health University of Brescia, ASST-Spedali Civili, 25123 Brescia, Italy; paolo.bossi@unibs.it (P.B.); a.alberti015@unibs.it (A.A.); 3U.O.C. Odontostomatologia, A.O.R.N. Cardarelli, 80131 Naples, Italy; a.guida@istitutotumori.na.it; 4Radiation Therapy Unit, INT IRCCS Foundation G Pascale, Via M. Semmola, 80131 Naples, Italy; p.muto@istitutotumori.na.it; 5Department of Abdominal Oncology, SSD-Innovative Therapies for Abdominal Cancers, Istituto Nazionale Tumori di Napoli, IRCCS “G. Pascale” Via M. Semmola, 80131 Naples, Italy; a.ottaiano@istitutotumori.na.it; 6Medical Oncology Unit, Azienda Ospedaliera S. Anna e S. Sebastiano, 81100 Caserta, Italy; davleo@inwind.it (D.L.); mrc.defelice@gmail.com (M.D.F.); 7Otolaryngology and Maxillo-Facial Surgery Unit, Ospedale Casa Sollievo della Sofferenza, 71013 San Giovanni Rotondo, Italy; f.longo@istitutotumori.na.it; 8Sarcoma and Rare Tumors Medical Oncology Unit, Istituto Nazionale Tumori di Napoli, IRCCS “G. Pascale” Via M. Semmola, 80131 Naples, Italy; s.tafuto@istitutotumori.na.it; 9Medical Oncology Unit Sir Apicella Hospital, ASL Na3 Sud, Pollena Trocchia, 80040 Naples, Italy; giuseppina.dellavittoria@gmail.com; 10Medical and Experimental Head and Neck Oncology Unit, INT IRCCS Foundation G Pascale, Via M. Semmola, 80131 Naples, Italy

**Keywords:** squamous cell carcinoma of the head and neck, recurrent/metastatic, multidisciplinary team management, abscopal effect, immunotherapy

## Abstract

**Simple Summary:**

The purpose of this manuscript is to illustrate the difficulties associated with the treatment of the patient with recurrent metastatic SCCHN. This type of patient is very heterogeneous, including very different cases both as regards the natural history and the types of treatment to be used. The authors then illustrated the possible therapeutic options available and tried to propose the best strategies to be adapted to the individual case, based on the characteristics of the patient and the disease. The main conclusion of the following work is that the multidisciplinary approach is the winning weapon in any patient.

**Abstract:**

Squamous cell carcinoma of the head and neck (SCCHN) is a complex group of malignancies, posing several challenges to treating physicians. Most patients are diagnosed with a locally advanced disease and treated with strategies integrating surgery, chemotherapy, and radiotherapy. About 50% of these patients will experience a recurrence of disease. Recurrent/metastatic SCCHN have poor prognosis with a median survival of about 12 months despite treatments. In the last years, the strategy to manage recurrent/metastatic SCCHN has profoundly evolved. Salvage treatments (surgery or re-irradiation) are commonly employed in patients suffering from locoregional recurrences and their role has gained more and more importance in the last years. Re-irradiation, using some particularly fractionating schedules, has the dual task of reducing the tumor mass and eliciting an immune response against cancer (abscopal effect). In this review, we will analyze the main systemic and/or locoregional strategies aimed at facing the recurrent/metastatic disease, underlining the enormous importance of the multidisciplinary approach in these types of patients.

## 1. Background

Squamous cell carcinoma of the head and neck (SCCHN) represents the sixth most common malignancy worldwide and its mean incidence rate is about 20 per 100,000 people in the regions of Europe, China, the Indian subcontinent, South America, and among African Americans in the United States. The incidence of SCCHN widely varies according to the areas of the globe, with these neoplasms being much more frequent in the Eastern countries rather than in the Western ones [1]. 

The well acknowledged risk factors are smoking, alcohol consumption, different forms of chewing tobacco, and chronic oral trauma. Lately, the infection sustained by Human Papilloma Virus (HPV) has been recognized as responsible for a fair percentage of SCCHNs, in particular those arising from oropharynx, whose incidence is steadily increasing in the last 10 years [2].

In the clinical practice, most patients with diagnosis of SCCHN have a locoregionally limited disease (from T1-2N0M0 to T4N3M0), and only 10% of them suffer from distant metastases. Upfront surgery followed by adjuvant radiotherapy or chemoradiotherapy, or in alternative exclusive chemoradiation (concomitant or sequential) are the current treatment options for locoregional disease. Even with recent advances in surgery and radiation, however, a subset of patients will eventually experience disease progression [3]. 

Mounting evidence highlight that, in patients diagnosed with locoregionally advanced SCCHN who have been treated with upfront surgery or upfront chemoradiation, the percent of locoregional failure is about 40–50%, while distant failure is 20–30% [4,5]. 

Thus, as a consequence, about a half of the patients with SCCHN have a recurrent/metastatic disease in common clinical practice. Recurrent/metastatic SCCHN is a difficult disease to treat, with poor prognosis and a median survival of about 12 months [6]. Several therapeutic strategies have been developed over the years, the monochemotherapy, the polychemotherapy, the association of chemotherapy plus cetuximab and, lately, the “new generation immunotherapy”, which exploits check-point inhibitors.

All the aforementioned strategies have a common feature, namely the idea that the recurrent/metastatic disease could be considered as a systemic disease, characterized by both macro and/or micrometastases disseminated in the blood and in the organs supplied by it. This is the reason for which the categories “recurrent” and “metastatic” SCCHN are grouped in the same prognostic category (recurrent/metastatic disease) and they share the same poor prognosis. Nevertheless, in some rare cases, particularly as regards patients with single locoregional relapse (recurrent SCCHN), surgery and/or radiotherapy are able to “cure” the patient, offering him an excellent chance of survival. Surgery and irradiation/re-irradiation are limited to few and selected cases, namely in patients with a single recurrence suitable for a locoregional treatment [3], and they should be evaluated as first option in these cases, being a strategy associated with longer overall survival. 

In conclusion, in recurrent/metastatic setting, locoregional strategies should be mainly aimed at debulking, symptom control and improving the quality of life, while, more rarely they can obtain a “cure” for the patient. As a matter of fact, there is no clear evidence regarding locoregional “curative” treatments (surgery/re-irradiation) capable of significantly improving survival. 

In this review we will analyze the different therapeutic options available in patients diagnosed with recurrent/metastatic SCCHN and using a systematic review of the data concerning various published clinical studies, we will propose a personalized treatment algorithm based on the characteristics of the patient and the disease. The aim of the work is to demonstrate that the best approach to the patient in the recurrent/metastatic setting is the multidisciplinary one.

## 2. Salvage Surgery in Patients with Recurrent/Metastatic Disease

Surgery and/or radiotherapy with or without concurrent chemotherapy has been established as a primary treatment for all newly diagnosed SCCHN. Nevertheless, loco(regional) failure occurs in up to 50% of patients, and for these lasts, salvage surgery is considered the best treatment option [3]. Even after aggressive multimodal therapy, many patients have persistent or recurrent disease due to their multiple risk factors and induced field cancerization. Salvage surgery represents the optimal approach in this category of patients, but it is not always easy to identify the ideal candidate for surgical approach, and moreover, salvage surgery may not be performed in all patients. With persistence/recurrence rates as high as 40% to 50%, and only 20% to 30% of these patients eligible for curative salvage therapy [7,8], it is clearly important to determine which factors predict successful salvage.

Matoscevic et al., performed a descriptive analysis of 176 patients with SCCHN, who relapsed after the primary curative treatment. Patients with a previous diagnosis of primaries arising from the oral cavity, larynx, oropharynx, and hypopharynx and who were treated with upfront surgery or in alternative radiation therapy were enrolled. Salvage surgery was the most frequently chosen modality of salvage treatment, being performed in 67.2% of patients affected by laryngeal cancer, 45.3% of those with oropharyngeal cancer, 56.5% of patients with hypopharyngeal carcinoma, and 37.8% of those affected by oral cavity cancer. The salvage rate was significantly better in patients affected by larynx and oral cavity recurrent tumors, while those suffering for oropharynx and hypopharynx recurrences had worser outcome. The authors also highlighted that lower initial T and N stage, developing local relapse in larynx and oral cavity have the best chances for salvage [9].

Elbers et al., used salvage surgery in 189 patients with a diagnosis of recurrent SCCHN carcinoma, previously treated with radiotherapy alone or chemoradiotherapy as an upfront approach. Oral cavity primaries were excluded from the analysis, upfront surgery being in this case the best option in site of radiation therapy. The authors found that larynx carcinomas were associated with more favorable local and locoregional control than pharyngeal (both oropharynx and hypopharynx) carcinomas. Overall, surgery reached a five-year overall survival (OS) of 33%, and a median OS of 18 months. Moreover, a sharp distinction was made between the “local” disease (recurrence on T) and “locoregional” disease (recurrence upon T and/or N), and interestingly, survival following salvage surgery for locoregional disease was significantly worse if compared with those obtained in local disease. In patients with locoregional recurrence, due to the higher tumor burden, prognosis was significantly worse. In this category of patients, the authors also found that rpT stage and ASA (American Society of Anesthetists) score were independent predictors for worse DFS [10].

Most data from the literature have highlighted that salvage surgery for laryngeal and oral cavity tumor recurrences does result in better survival, if compared to other subsites (hypopharynx and oropharynx), and this may be due to the major difficulty in resecting pharyngeal recurrences due to nearby anatomical structures (e.g., carotid artery and skull base) [8,9,10]. Nevertheless, salvage surgery may be taken into account, especially in high expertise centers, as modality of choice also in recurrent pharyngeal cancers. When choosing surgery as salvage strategy is important to acknowledge some important features, mainly the site of recurrence and then the extent of recurrent disease (pT), the ASA score, the disease-free interval, the feasibility of postoperative radiation, and the previous radiotherapy received, the performance status of the patient and the family and social environment.

## 3. The Evolving Role of Re-Irradiation (Curative or Palliative Intent)

### 3.1. Introduction

A number of patients with locally advanced disease who have been treated with radiation therapy will experience a recurrence [11,12]. Radiation therapy could also be used in the latter, but the total dose delivered as an upfront strategy strongly conditions the remaining deliverable dose in the “recurrent/metastatic” setting.

### 3.2. RT Schemes Employed in the Clinical Practice for Recurrent/Metastatic Disease

The anatomical structures of the head and neck area, having already received a high dose of radiation, cannot be further subjected to treatments involving excessive doses. The anatomical structure on which the radiation oncologist’s attention is most focused is the spinal cord of the cervical tract, which often after an adjuvant or exclusive treatment at a variable dose between 64 and 70 Gy, has already absorbed a medium-high dose of radiation (about 25–35 Gy). The main consequence is that re-irradiation is not always feasible in these patients [13,14,15].

Re-irradiation in recurrent/metastatic disease is an intriguing option and some authors have demonstrated that it can improve both activity and quality of life of patients [16,17,18]. Salvage surgery has proven to be the most effective curative-intent treatment and it is the treatment of choice for all patients with resectable tumors and sufficiently good health status. However, increasingly growing evidence [16] pushes current guidelines to consider both salvage surgery and irradiation/re-irradiation almost on the same level, for the treatment of recurrent/metastatic SCCHN, although head-to-head studies have not been performed. 

In some cases, the re-irradiation can have a “curative” purpose (single or oligometastatic disease), while in others it is simply palliative. The maximum dose delivered and the technique used can also change profoundly. However, even palliative treatment is accompanied by clinical benefit; in fact, some authors, using the results from systematic reviews and expert panel-generated recommendations, carried out a decision analysis model which showed that concurrent chemotherapy and reirradiation, albeit palliative, offer a significant improvement in quality of life (QOL), if compared to the best supportive care [19]. 

### 3.3. Palliative Schemes

Most trials exploring the role of re-irradiation in unresectable recurrent/metastatic SCCHN are retrospective studies, but the data obtained from their careful analysis revealed that at 2 years, one quarter to one third of the patients will be free of locoregional recurrences. Nevertheless, late severe (G3–G4) toxicities occur in up to 40% of reirradiated patients, and nearly 10% of patients will have treatment-related deaths [20,21,22,23,24]. In fact, an important point to consider is that re-treatment is associated with an increased risk of serious toxicity and impaired quality of life.


Widely used protocols provide for the administration of 20 Gy for palliative purposes, but also alternative schemes (both conventional and altered fractionation) can be used depending on the time elapsed since the last RT, the dosage used, and the response obtained. 

Radiation therapy using a dose of 20 Gy divided into five fractions, in some clinical trials, results in an overall response rate (ORR, i.e., the sum of complete responses and partial responses) of 30–37% and a fairly good symptom relief [25]. Alternative schemes are the split courses that are composed of two cycles of 25 Gy in 10 fractions, reaching a total of 50 Gy. The reported data regarding both tumor response and symptom relief seem to be better than those observed with 20 Gy [26]. Another interesting scheme is the so called “Quad Shot regimen”, which comprises three courses of bis in die (BID, meaning twice a day), giving 3.7 Gy per fraction for two successive days to achieve a total of 44.4 Gy in 12 fractions. The Quad Shot regimen has reached an ORR ranging from 50% to 70% in clinical trials and an impressive quality of life improvement [27]. Porceddu et al., using a regimen of 30–36 Gy in five to six twice-weekly fractions, described in their analysis an ORR of 80%, while Agarwal et al. with an intensive palliative RT regimen delivering 40 Gy in 16 fractions, obtained an ORR of 73% and symptom relief of 75% [28]. 

### 3.4. Curative Schemes

The possibility of conforming the radiation beams to the target volume, with the possibility of considerably reducing the dose delivered to the organs at risk, has also made it possible, over the years, to re-irradiate with maximum doses up to 66–70 Gy. Both the modulation of beam intensity and the possibility to perform the image guidance have been largely adopted in the RT clinical practice, due to their potential to significantly change the toxicity profile and/or treatment efficacy, if compared with the conventional RT. These are the reasons for which IMRT (intensity modulated radiation therapy) and SBRT (stereotactic body radiation therapy) are potentially better re-irradiation techniques than conventional RT. Data regarding the comparison between IMRT/SBRT and conventional techniques can be drawn mostly by retrospective trials and overall, there are no differences in terms of survival. On the other hand, improvement in local tumor control can be seen [16,29,30]. An aspect to be clarified regarding the IMRT and SBRT techniques, concerns their toxicity. In fact, these techniques allow for the minimization of the dose delivered to the spinal cord and therefore the risk of transverse myelitis does not represent a big problem even in re-irradiation treatments. On the other hand, the most feared side effects with such treatments are severe dysphagia and bone necrosis [29,30].

Rühle et al. treated 48 patients with diagnosis of recurrent or second primary SCCHN at the University of Freiburg Medical Center. The medium dose reached on the clinical target volume was 58.4 Gy, at a price of a moderate acute/late toxicity. In fact, 10.4% of patients suffered from at least one grade 3 toxicities, while 27.3% of experienced chronic higher-grade toxicities (≥grade 3) with 3% grade 4 carotid blowout and 3.0% grade 4 osteoradionecrosis [31].

Caudell et al. [32] performed a multi-institution retrospective cohort study to investigate the effect of the elective treatment volume, dose, and fractionation on outcomes and toxicity in patients with recurrent, previously irradiated SCCHN [32]. Five hundred and five (505) patients were included in this analysis. As result, the median dose of re-IMRT was 60 Gy (range 39.6–79.2) and the rate of overall late grade ≥3 toxicity was 16.7%. 


Briefly, data concerning the RT technique to be used in the clinical practice are conflicting and, presently, no clear recommendations can be made. Generally, where possible, it is possible to reach a maximum dose close to 66–70 Gy, in patients in whom a rapid and effective debulking (single unresectable locoregional relapse) is desired. However, palliative treatments (20–30 Gy) are often considered in patients with multiple district relapse or with compromised Performance Status. 

One aspect to be clarified, however, concerns the purpose of the re-irradiation treatment. In fact, even the so-called “curative treatments” are aimed mainly at debulking and symptom control, although a small percentage of patients can achieve disease recovery and a long disease-free interval [32].

### 3.5. The Abscopal Effect

The therapeutic effect of RT is mediated not only through the release of energy by the ionizing radiations directly into the tumor tissue, but also by the so-called abscopal effect, wherein the distant lesions respond to locoregional treatment. The term “abscopal” means “to go beyond the scope” and it occurs when a reduction in disease is seen outside the irradiated site [33,34].

RT can damage cellular macromolecules, particularly the DNA, effectively inducing growth arrest and cell death. In addition to this well acknowledged mechanism of action, RT also exerts immunostimulant effects on the host. In fact, RT modulates the immune response against cancer by inducing “immunogenic cell death” (ICD), with release of tumor-associated antigens (TAAs) [35].

ICD provokes the tumor cell’s release of the so-called DAMPs (damage associated intra-cellular pathways), namely proteins and/or small molecules able to induce and stimulate the maturation of dendritic cells (DCs) and their conversion to antigen presenting cells (APC). The outcome is the activation of the innate and adaptive immune systems. Examples of DAMPs are calreticulin, high-mobility group box 1 protein, and adenosine triphosphate (ATP) [36,37,38]. Another mechanism of immune response induction is through IFN secretion. In fact, cytosolic DNA derived from irradiated tumor cells may be taken up by APCs. Once captured it is recognized by cyclic-GMP-AMP synthase which in turn can induce the nuclear translocation of stimulator of interferon genes (STING). This last becomes capable to recruit and phosphorylates TANK-binding kinase 1, which finally activates interferon regulatory factor 3 (IRF3). Activated IRF3 leads to the expression of type I IFN. IFN stimulates CXCL9 and CXCL10 expression, which recruit CXCR3-expressing T cells into the tumor microenvironment (TME). The final result is the activation of APCs which become able to take up antigens, and more specifically to present TAAs to cytotoxic T-lymphocytes [39,40,41]. 

On the other hand, there are data favoring an immunosuppressive effect of RT on the TME (tumor microenvironment). The TME can be seen as the point of contact between the tumor cells and the host immune system. The effects of RT on the TME can modify the balance from an immunosuppressive (also called “cold”) TME to an immunostimulatory (“hot”) TME [42,43,44]. The so called cold TME is characterized by a low number of T-Cytotoxic lymphocytes, Natural Killers and APCs, and simultaneously, a high number of T-reg lymphocytes and myeloid derived suppress cells (MDSCs). The transition to a hot or cold TME depends on the cytokines produced in response to RT and, according to several data, the kind of cytokines released are strictly associated to the total and the fractionating dose of RT delivered [45,46].

Based on these findings, high doses of RT seem to favor the production of pro-tumorigenic (immunosuppressive) cytokines, such as TGF-beta and Hypoxia-inducible factor alpha (HIF-Alpha). The first provokes the cytotoxic-T-cells anergy, and so, the T-Reg expansion. HIF-Alpha, on the other hand, promotes a robust pro-inflammatory reaction in stroma, inducing IL-1, IL-6, IL-10 and TGF-beta release [47,48]. As opposed to it, low doses of RT, availing of the above mentioned mechanisms (DAMPs and STING pathway), induce the reversal outcomes, promoting APC and cytotoxic T-cell maturation [47,49].

After consulting the treatment plans related to previous radiation treatments and in particular, after evaluating the dose-volume histograms, the re-irradiation schedule should be chosen with the aim to increase both the local and the “at distance” control of disease. Well defined total doses and fractionating doses are related to the appearance of the abscopal effect.

Investigators have tested different total doses and fractionation schemes in a variety of pre-clinical models to maximize the abscopal effect and the results suggest that there is an optimal range (typically high dose per fraction) for the abscopal effect. More in particular, a dose per fraction close to 10 Gy with 1–3 fractions is likely optimal for abscopal effect induction. As a consequence, the best RT schedule seems to be the SBRT, followed by hypofractionated treatments [26,27,28,47]. First combinations of check-point inhibitors added on radiation therapy and chemotherapy did not confirm the theoretical rationale of a synergistic effect. Data from other trials are awaited in the next few years that will clarify the precise role and the best possible combination of immunotherapy and RT in locally advanced SCCHN.” ADD REFERENCE: “910O—Cohen EE, Ferris RL, Psyrri A, et al. Primary results of the phase III JAVELIN head & neck 100 trial: Avelumab plus chemoradiotherapy (CRT) followed by avelumab maintenance vs CRT in patients with locally advanced squamous cell carcinoma of the head and neck (LA SCCHN). ESMO Virtual Congress 2020”. 

Despite all these intriguing and clear premises, the first published results of the combination between immunotherapy and radiation in recurrent/metastatic SCCHN have been unsatisfactory regarding abscopal effect. The combination of RT and ICI can, however, be promising [50,51].

## 4. Post-Operative Re-Irradiation (+/− Chemotherapy)

Salvage surgery is, where feasible, the standard option in patients with recurrent SCCHN. However, the failure rate of this therapy is still too high and for this reason it is important to use additional therapies capable of optimizing the effectiveness of salvage surgery. Adjuvant re-irradiation has always been considered an intriguing technique and therefore widely used in clinical practice. Janot et al. [52] carried out a prospective clinical trial enrolling patients with recurrent SCCHN previously irradiated and, once in relapse, treated with salvage surgery. These patients were randomly assigned to observation or concomitant chemo-reirradiation. Sixty-five patients were enrolled, and the RT technique used wea the conformal 3D, reaching a medium total dose of 60 Gy. Hydroxyurea and 5fluorouracil were the chemotherapeutic agents administered concomitantly with radiotherapy. At 2 years, 39% of patients in the “chemo-radiotherapy arm” and 10% in the “observation arm” experienced grade 3 or 4 late toxicity according to Radiation Therapy Oncology Group criteria. Disease-free survival (DFS) was significantly improved in the “chemo-radiotherapy arm”, but OS was not statistically different. The authors concluded that the addition of radiotherapy to salvage surgery can improve local disease control at the cost of considerable toxicity. So, patients need to be better selected for adjuvant treatment after salvage surgery. For this purpose, some unfavorable prognostic factors, such as extracapsular lymph node invasion, infiltration of the resection margins and perineural infiltration, have been proposed as parameters capable of selecting patients with worse prognosis and therefore deserving of radiotherapy (+/− chemotherapy) after salvage surgery. Vargo et al. [53] retrospectively reviewed data regarding 28 patients with recurrent SCCHN previously irradiated and later treated with salvage surgery. These patients had several poor prognostic factors, such as positive surgical margins and/or extranodal extension, and they received adjuvant SBRT ± cetuximab. SBRT consisted of 40 to 44 Gy in five fractions over 1 to 2 weeks with concurrent cetuximab administered at 400 mg/m^2^ induction dose, followed by weekly 250 mg/m^2^ of maintenance dose. The 1-year locoregional control, distant control, disease-free survival, and overall survival were 51%, 90%, 49%, and 64%, respectively. The rates of acute and late severe (≥grade 3) toxicity were 0% and 8%, respectively. The authors concluded that the treatment was effective and not very toxic, paving the way for a possible use of cetuximab in place of chemotherapy in this disease setting. Takiar et al. [54] retrospectively reviewed the data of 227 patients who received head and neck reirradiation using IMRT from 1999 to 2014. Two hundred and six (206) patients were treated with “curative” IMRT. Fifty percent (50%) underwent salvage surgery followed by IMRT (+/− chemotherapy) and the remaining part underwent definitive IMRT (+/− chemotherapy). The re-irradiation dose was higher for definitive (median 66 Gy) versus adjuvant (median 60 Gy) IMRT patients (*p* < 0.01) Concomitant cisplatin administered every three weeks at a dose of 100 mg per square meter of body surface, was the most employed drug. Five-year locoregional control, progression-free survival, and overall survival rates were 53%, 22%, and 32%, respectively. On multivariate analysis, concurrent chemotherapy was associated with tumor control, whereas performance status was associated with survival. The authors concluded that IMRT either definitively or after salvage surgery can produce promising local control and survival, but treatment-related toxicity remained significant.

In conclusion, re-irradiation can be used in the adjuvant phase, after salvage surgery, taking into account that the related toxicity is not negligible and that a patient selection is therefore necessary on the basis of the risk of relapse. The presence of infiltration of the resection margins and/or extracapsular lymph node metastases is an indication to carry out this treatment. The technique to be used should be IMRT/SBRT and where possible it should be combined with concomitant chemo or cetuximab.

## 5. The Role of Electrochemotherapy

Electrochemotherapy (ECT) is a methodology able to couple the “electroporation” of the cell membranes with the concomitant administration of antineoplastic drugs. Electroporation consists of the application of short-intensity pulsed electric fields to tumor cells, following which, the plasma membrane permeability to different hydrophilic drugs transiently increases, thus facilitating cellular uptake of cytotoxic agents [55,56,57,58]. ECT acts through three main mechanisms, namely, the so called “vascular lock”, which consists of a vascular spasm able to interrupt the tumor bleeding and to a prolong contact time between the antineoplastic drug and the tumor tissues; the second mechanism is the direct action of the drug with the cancer cells DNA, which leads to DNA damage and quick apoptosis; finally, the most interesting one regards the induction of the “immunogenic cell death” (ICD), which is characterized by the tissue expression of different immunogenic antigens and a strong recruitment of DC and APC. ECT, just like the SBRT can induce the DAMPs in the tumor cells, leading so to the APC and finally the cytotoxic T-lymphocytes. This consideration paved the way to different clinical trials combining ECT with last generation immunotherapy, with the aim to reinforce the abscopal effect elicited by ECT [58,59,60].

Beyond the likely abscopal and immunological effect induced by ECT, this methodology is currently and largely employed in the clinical practice and it is part of the management of patients with symptomatic, inoperable, chemo-, and radioresistant lesions. Longo et al., treated 93 patients with recurrent and/or metastatic head and neck tumors heavily pretreated with one or more chemotherapy lines, with bleomycin based ECT. The main endpoints of the trial were quality of life, pain, and bleeding control, while the secondary endpoints were ORR and disease control rate (DCR, i.e., the sum of the stable disease, partial responses and complete responses). As results, an optimal control of bleeding and pain were achieved, and more interestingly, a DCR rate of 79% was seen [61]. In a phase II trial, Plaschke et al. treated 36 patients with recurrent/metastatic SCCHN with the same ECT regimen (bleomycin based), and as results, an ORR rate of 56% was reached [62]. Very similar results were achieved also by Bonadies et al., in their phase II prospective trial, in which 26 patients with SCCHN received ECT and 58% of them experienced a response (partial or complete) [63]. 

The main indication for ECT in the SCCHN management regards the “palliative setting”, in combination with systemic treatments and in patients judged to be inoperable and/or not re-irradiable. Due to its immunologic properties, ECT should be in a near future, employed in combination with immunotherapy.

## 6. The Role of Systemic Strategies (Chemo and Immunotherapy)

In most cases, systemic therapies are the only choice for patients with recurrent/metastatic disease. Combination of Cisplatin and 5Fluorouracil has represented the preferred regimen for several years, yielding an overall survival (OS) of about 9 months in clinical trials. As alternative, in patients unfit for the polychemotherapy, single agent cisplatin or methotrexate were extensively used in the clinical practice, nevertheless obtaining a significantly lower OS [64,65,66,67].

Vermorken et al. published the results of a seminal clinical trials, the EXTREME-trial (Erbitux in First-Line Treatment of Recurrent or Metastatic Head and Neck Cancer), demonstrating that the addition of the anti-EGFR monoclonal antibody cetuximab to the standard cisplatin-5fluorouracil significantly improved the OS in patients with recurrent/metastatic SCCHN. The new standard Cetuximab-cisplatin-5Fluorouracil obtained an OS superior to 10 months, becoming so the new first-line standard regimen to be used in these patients [68]. 

Recently, immunotherapy has quickly entered solid tumors clinical practice. Immunotherapy is a strategy able to reactivate and reinforce the immune response of the host against tumor cells. The rationale upon which it is based is the existence of the Tumor Associated Antigens (TAA), namely proteins produced by the tumor cells, which are able to elicit an immune response. The result is tumor rejection. Nevertheless, immune response against TAA is rarely elicited and it often fails, thus leading to tumor growth and progression. The mechanisms underlying this failure are to be found in the ability of tumor cells to depress immune response through the production of immunosuppressive cytokines and through the recruitment of immune cells capable of attenuating the cell-mediated antitumor response [69]. The new generation immunotherapy is a strategy capable of eliminating these “immunological brakes” and reactivating the cell-mediated immune response directed towards TAA. 

Thus, nivolumab, a monoclonal antibody recognizing PD-1 (programmed death 1), in the phase III clinical trial called CheckMate141, was shown to improve prognosis over the physician’s choice of treatment in patients who had progressed to platinum-based chemotherapy with or without cetuximab. Nivolumab has reached an ORR (the sum of partial responses and complete responses) of 13% in a poor prognosis category of patients (resistant to platinum), also prolonging the OS (7.7 months vs. 5.1 months in the control arm) at a price of a fairly good toxicity spectrum. For these reasons, nivolumab is commonly employed in clinical practice [70].

While nivolumab has reached approval as treatment in platinum refractory patients, more recently another PD-1 inhibitor has been tested in patients with first line recurrent/metastatic SCCHN, with no resistance to platinum-based therapy. So, in the Keynote-048 trial, pembrolizumab associated or not with platinum plus 5Fluorouracil was compared with the standard platinum plus 5Fluorouracil plus cetuximab. As results, pembrolizumab alone achieved a significantly higher OS if compared with standard chemotherapy (14.9 vs. 10.7 months, *p* < 0.0086) in patients whose tumor over-expresses tissue PDL-1 with a Combined Positive Score, CPS ≥ 20), but no difference in survival was shown in the CPS 1–19 subpopulation [71].The association of pembrolizumab and chemotherapy was better than standard chemotherapy (13.0 vs. 10.9 months, *p* < 0.03) in the Intent To Treat (ITT) population (independently from the PDL-1 expression). These results have led regulatory authorities to approve pembrolizumab as first-line therapy in recurrent/metastatic SCCHN, added or not to platinum plus 5Fluorouracil [72].

Although pembrolizumab and nivolumab have been shown to significantly improve OS and PFS (progression free survival) if compared with the control arms, respectively, as a first and second line of treatment, their activity in terms of response when employed alone has not been particularly impressive. In fact, nivolumab showed a 13% ORR and pembrolizumab a 17% ORR when considered in the total population and a 23% ORR in the population with PDL-1 CPS ≥ 20 [70,71]. When pembrolizumab is given as first line with chemotherapy, the ORR in the total population did not differ from what obtained in the platinum-5Fluorouracil plus cetuximab arm (36%), highlighting that check-point inhibitors improve survival over standard therapy increasing duration of response and not response rate. Indeed, obtaining a response is crucial in SCCHN, where the presence of masses that compress vital structures located in the neck (carotid, trachea, and vascular-nerve bundle) necessitates rapid debulking of the disease and where the global improvement is linked to reduction of disease burden [73].

In this regard, the possibility to obtain a rapid tumor shrinkage, which could significantly improve the symptoms and quality of life of patients should be one of the treatment’s aims. The integration of other therapeutic opportunities, such as palliative surgery, radiation or reirradiation and electrochemotherapy should be evaluated in the multidisciplinary team.

## 7. The Role of Metastasis Directed Therapy (MTD)

The recurrent/metastatic setting is very heterogeneous, including patients with different “tumor burdens”. Some patients have a high number of metastases, while others have an oligo-metastatic disease (total number of metastases less than or equal to 3). Some authors have focused their attention on the genetic evolution of metastatic diseases, underlining how it can be very heterogeneous from disease to disease. From this it follows that some diseases, although metastatic (olIgo-metastatic), tend to have a less aggressive clinical history and therefore metastasis directed therapy (MTD) can be used in them [74]. 

Beckam et al. [75] performed a descriptive analysis of 186 patients with diagnosis of recurrent/metastatic SCCHN who relapsed after primary treatment. The authors described the outcome of these patients (OS and DFS) also relating it to the total number of metastatic sites found on each patient. As result, they discovered that patients with a single metastasis had a 5-year OS of 35% in contrast to patients with multiple metastases with a 5-year OS of 4% and the difference was statistically significant. The authors concluded that patients with limited metastatic disease may derive significant benefit from MDT. 

Shiono et al. [76] carried out a very similar retrospective analysis of 114 patients diagnosed with SCCHN lung metastases who underwent one or more lung metastasectomies. The mean tumor size was 3.3 cm (range, 0.7 to 11 cm), while the median number of resected metastatic lesions per patient was 1 lesion (range, 1 to 6). The overall 5-year survival rate after pulmonary metastasectomy was 26.5%, and the median survival time was 26 months. Interestingly, the number of resected metastases did not statistically significantly impact overall survival, suggesting that even multiple resections, provided they are radical, can be proposed in clinical practice.

In the future, prospective randomized studies, associated with a determination of the genetics of the disease (oligometastatic versus multi-metastatic) could support the multidisciplinary team in deciding which patients to candidate for MTD.

## 8. Discussion and Conclusions

SCCHN account for 7% of cancer cases and about two-thirds of these patients present with locally advanced disease in which, despite aggressive multimodal treatment, relapses occur in around 50% of cases [1,3]. In patients with recurrent and/or metastatic SCCHN, the most effective systemic treatment offered a median overall survival of 10 months in the pre-immunotherapy era. Lately, the results of the Keynote-048 trial have changed the standard of systemic therapy and, interestingly, have provided us with a new standard of systemic therapy for tumors expressing PDL-1, namely the pembrolizumab-based therapy. The aforementioned results paved the way to a “new era” characterized by the affirmation of checkpoint inhibitors in clinical practice [71]. Also, nivolumab, in the setting of platinum refractory recurrent SCCHN is considered an “interesting weapon”, both in terms of efficacy and activity [67].

Nevertheless, “it is not gold all that glitters” and the median proportion of patients responding to immunotherapy is below 20% [70,71]. In addition, about 60% of SCCHN patients receiving immune checkpoint inhibitors, experience immune-related adverse effects, and in 17–18% of patients, these are grade 3 or higher. As said, ORR in patients treated with immunotherapy is not satisfactory and this last may be a very significant matter, particularly in patients affected by wide and very symptomatic neck massess. 

For the above mentioned plus other reasons, the management of recurrent/metastatic SCCHN has lately evolved and it is now aimed to reach different goals. In fact, other than the efficacy, also the activity of the chosen treatments should be taken into account when planning a treatment, and strategies aimed to reinforce or coadiuvate systemic therapies could be always considered. Salvage surgery and/or re-irradiation are considered the “gold standard” in patients suitable for a salvage treatment, namely in patients with a local/locoregional recurrence who are fit for a local treatment. In these patients, salvage treatments may also allow to a fairly good outcome. Nevertheless, both surgery and re-irradiation are burdened by not insignificant toxicities, and importantly, by a high rate of second and third recurrences [77,78]. Therefore, research on the possible means of enhancing the anti-PD-1 effect, and overall, the systemic therapy effect, is much needed. 

Radiation therapy (RT) is acknowledged as a locoregional strategy and its effects are known to be mediated by loss of tumor cells’ reproductive ability, other than their apoptosis induced by massive DNA damage. Now, the complex interaction between the immune system and RT is well understood, and interestingly, there are lots of data highlighting that reduction of tumor burden is dependent on functional T cells, even after ablative RT doses [79,80,81]. RT in fact, acts as in situ vaccination, so it can promote tumor antigen cross-presentation and to induce the production of cytokines which, at the end, enhance the local and abscopal antitumor immune responses. For a long time, the RT net effect on the immune system was understood as immunosuppressive, but it is now clear that the final effect on the immune response may depend on the fraction dose and on the total dose erogated, since SBRT and hypofractionated regimens are more associated with an immunostimulating effect. The above considerations lead to the concurrent use of RT and immunotherapy, when feasible, in patients with recurrent/metastatic SCCHN, especially in presence of wide and highly symptomatic masses which need of a quick debulking.

A similar immunostimulant effect may be done by electrochemotherapy (ECT), which is, in the same way as RT, considered to be a locoregional treatment [58]. ECT is commonly used as palliative treatment in patients who need a rapid response on symptomatic lesions, namely very painful and/or bleeding masses. ECT can easily enter the therapeutic armamentarium provided for SCCHNs, since it is quick and easy to perform. Its combination with immunotherapy could be a very intriguing solution.

To conclude, recurrent/metastatic SCCHN are a big challenge for the oncologists, but their management can be greatly facilitated by the multidisciplinary approach. The “right way” to take does not translate into a single strategy, but rather from the perfectly synchronized use of multiple therapeutic modalities, adapted on the basis of the patient/tumor characteristics. A scheme indicating our purpose to manage recurrent/metastatic SCCHN is depicted in Figure 1.

## Figures and Tables

**Figure 1 cancers-13-02371-f001:**
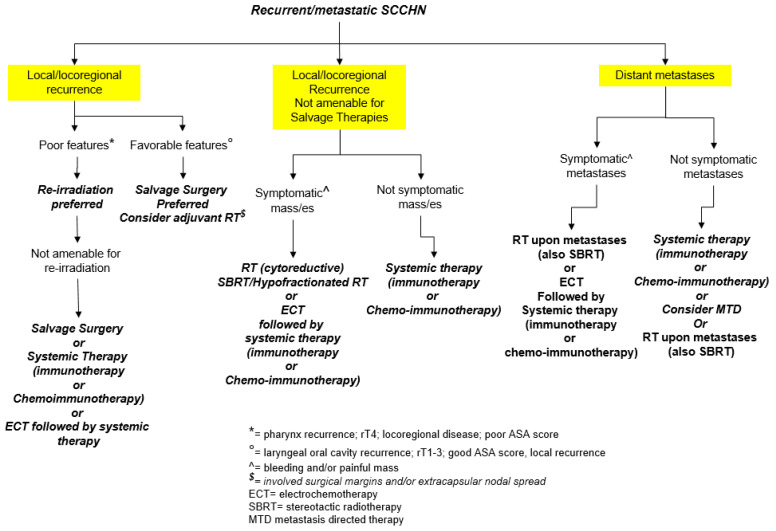
A flow-chart illustrating the therapeutic options in recurrent/metastatic SCCHN.

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
