# Peer review of "Recurrent/Metastatic Squamous Cell Carcinoma of the Head and Neck: A Big and Intriguing Challenge Which May Be Resolved by Integrated Treatments Combining Locoregional and Systemic Therapies"

_cancers, 2021, doi:10.3390/cancers13102371_

Round 1

Reviewer 1 Report

Thanks for allowing me to review this manuscript. The authors have performed a review to summarize knowledge about treatment of recurrent head and neck cancer. This is truly a challenging topic to write about, since it is a very heterogenous group.

The authors have done a great job tracking down current and up to date information, but in its current format it is hard to read as there is significant mix of modalities in the different subheadings. I think this is worthwhile publishing, but requires significant restructuring in how it is presented.

The introduction needs to be much more brief, just presenting the topic and giving a succint intro. Towards the end it starts presenting data on different modalities of immunotherapy. I would save this for a different subheading. Also I object to the following sentences in the introduction.

"All the aforementioned strategies have a common feature, namely the idea that the recurrent/metastatic disease can be considered as asystemic disease, characterized by both macro and/or micrometastases disseminated in the blood and in the organs supplied by it. This is the reason for which the categories“recurrent”and “metastatic”SCCHN are grouped inthe same prognostic category(recurrent/metastatic disease)and they sharethe same poor prognosis.Surgery and irradiation/re-irradiation arelimited to few and selected cases, namely in patients with a single recurrence suitable for a locoregional treatment, and they should be evalautaed as first option in these cases, being a strategyassociated with longer overall survival"

If recurrent disease is considered systemic, why do we offer surgery at times..?

I would try to reorganize the review in the following subheadings.

1.1Role of salvage surgery in previous (C)RT alone

1.2Role of salvage surgery after surgery and (C)RT

2.1Role of curative Re-irradiation either with or without surgery

2.2Role of palliative radiation

3.0Role of other treatment strategies (ECT) or others

4.0Role of systemic strategies (chemo and immunotherapy)

5.0I'd also consider separating HPV related recurrent/metastatic disease and present outcomes and strategies separately.

I really think the authors have done a valuable review of the topic, but I think a reorganization of the presentation would make this review much more valuable for the readership.

Author Response

Dear Editor, please find attached the requested file

Best Regards

Dr Francesco Perri

Reviewer 2 Report

The authors present a detailed review regarding therapeutic options for the difficult clinical scenario of recurrent head and neck cancer. Overall, I feel the review was well-written and informative. I have only a few comments.

1) The authors correctly state that not all patients with recurrent HNC are candidates for radiation, often secondary to recurrent tumor volumes, disease-free interval, and dose to integral structures during initial treatment. However, reirradiation to definitive doses (66-70Gy) has been frequently described and is among the recommended treatments listed in the National Comprehensive Cancer Network guideline. As other fractionation schemes are detailed, doses in the definitive range are noticeably absent. I recommend this be briefly addressed. 

2) Please correct typos found in lines 200 (tecnique), 233 (mieloid), 254 (SRBT) and within the figure (SRBT)

Author Response

Dear Editor,

please, find attached the requested file

Best Regards

Dr Francesco Perri

Reviewer 3 Report

The authors present a comprehensive review on treatment modalities for recurrent/metastatic head and neck cancer, mainly focussing on re-irradiation, electrochemotherapy and salvage surgery and less on systemic therapy.

The topic is very interesting, the English language used fine and the most important current literature largely represented here.

However, I have some major remarks about points that have to be improved:

1) The title of the manuscript does not focus on local treatments and although you refer to the most important studies on chemo- and immunotherapy, this is clerarly not the focus of your manuscript. I would strongly recommend to either change the title or add n additional section (and the according discussion) about developements and challenges of the systemic treatment for these tumors.

2) The structure of the manuscript needs improvement: The "background" is clearly to long and the the first two paragraphs of the discussion are an almost 1:1 repetition of the information found under "background".

3) The authors have to make the goal of the manuscript more clear in the first section (as in the title)

4) The radiotherapy section appears also somewhat chaotic, mixing up the indication for salvage and strictly palliative treatment. Please separate these two aspects.

5) There is nor reference to radiotherapy doses needed for salvage treatment, or concomitant systemic treatments used (chemo-/anti-EGFR-/immunotherapy)

6) lines 153-154: sparing the spinal cord should be no problem with modern IMRT/SBRT. Please rephrase. Here you should also distinguish between salvage and palliative treatment regarding the dose and the risks and consider other risks like dysphagia and necrosis more than spinal cord injury.

7) lines 256-258: This should be rephrased/edited: "the study results were unsatisfactory regarding abscopal effect." The combination of RT and ICI can however be promising (e.g. Altay-Langguth et al., ctRO, 2021)

8) What happens in case of R+ or ECE-resection? what about postoperative Re-RT? It remains the only Re-RT indication supported with good, randomized data (Janot et al., J Clin Oncol, 2008)

9) Figure 1 should be changed and better discussed: Patients with poor features like organ dysfunction and low performance status are no good candiadates for Re-RT. There exists a lot of literature about that that should be considered (e.g. Tanventyanon et al., J Clin Oncol, 2009; Dornoff et al., Strahlenther Onkol, 2015). Furthermore, add the case of postoperative RT. SBRT is not really used as strictly palliative treatment as you present it here, but for treatment of distant  (oligo)metastases.

10) I totally miss the local/radical treatment of distant metastases in your manuscript (e.g. Beckham et al., BJC, 2019). If you refer to metastatic and not only locally recurrent disease in your title you should include this possibility.

Minor remarks:

Replace "tecniques" with "techniques" throughout the manuscript

Replace "primitives" with "primaries"

Figure legend: "SBRT = stereotactic radiotherapy"

line 265: some problem with the references

lines 325-329: is this the same statement in two sentences?

line 341: a space is missing

Author Response

(The authors gave the same response as above.)

Round 2

Reviewer 1 Report

The authors have done a great job re-organizing the manuscript and in its current format is much easier to read and understand.

Still, I completely disagree with the authors suggestion to group metastatic and recurrent disease together. How do the authors re-concile the following All the aforementioned strategies have a common feature, namely the idea that the 68recurrent/metastatic disease couldbe considered as asystemic disease, characterized by 69both macro and/or micrometastases disseminated in the blood and in the organs supplied 70by it. This is the reason for which the categories“recurrentand metastatic”SCCHN are 71grouped inthe same prognostic category(recurrent/metastatic disease)and they sharethe 72same poor prognosis.Surgery and irradiation/re-irradiation arelimited to few and se-73lected cases, namely in patients with a single recurrence suitable for a locoregional treat-74ment3, and they should be evalautaed as first option in these cases, being a strategyasso-75ciated with longer overall survival. In any case, the "recurrent / metastatic" setting is al-76most universally considered incurable, and any locoregional strategy should be consid-77ered palliative or in any case aimed at debulking, symptom control and improving the 78quality of life.In fact, there is no clear evidence regarding locoregional "curative" treat-79ments (surgery / re-irradiation) capable of significantly improving survival. 

Are the authors suggesting that recurrent disease is almost universally incurable?? I agree that the prognosis is poor, but clearly not incurable with the goal of debulking.

Surely metastatic disease is almost universally incurable, but not recurrent. I still strongly oppose the idea bunching these together. I have never seen this distinction anywhere else except in the setting of unresectable disease.

Author Response

Dear Editor, 

we analyzed the reviewer's advice and decided to modify our statements according to his suggestions

Reviewer 3 Report

The authors addressed all of the remarks raised by the reviewers and the manuscript clearly improved

Author Response

Dear editor,

we have performed the requested corrections

This manuscript is a resubmission of an earlier submission. The following is a list of the peer review reports and author responses from that submission.